ecology/evolution/microbiology

mechanosensory stimulus, decision-making, acetic acid bacteria, *Gluconobacter*, *Acetobacter*, spotted-wing *Drosophila*

**Authors for correspondence:**
Joanne Y. Yew
e-mail: jyew@hawaii.edu
Aya Takahashi
e-mail: ayat@tmu.ac.jp

# *Drosophila suzukii* avoidance of microbes in oviposition choice

Airi Sato[1], Kentaro M. Tanaka[1], Joanne Y. Yew[3] and Aya Takahashi[1,2]

[1]Department of Biological Sciences, and [2]Research Center for Genomics and Bioinformatics, Tokyo Metropolitan University, 1-1 Minamiosawa, Hachioji 192-0397, Japan
[3]Pacific Biosciences Research Center, University of Hawai'i at Mānoa, 1993 East West Road, Honolulu, HI 96822, USA

JYY, 0000-0003-4851-912X; AT, 0000-0002-8391-7417

While the majority of *Drosophila* species lays eggs onto fermented fruits, females of *Drosophila suzukii* pierce the skin and lay eggs into ripening fruits using their serrated ovipositors. The changes of oviposition site preference must have accompanied this niche exploitation. In this study, we established an oviposition assay to investigate the effects of commensal microbes deposited by conspecific and heterospecific individuals and showed that the presence of microbes on the oviposition substrate enhances egg laying of *Drosophila melanogaster* and *Drosophila biarmipes*, but discourages that of *D. suzukii*. This result suggests that a drastic change has taken place in the lineage leading to *D. suzukii* in how females respond to chemical cues produced by microbes. We also found that hardness of the substrate, resembling that of either ripening or damaged and fermenting fruits, affects the response to microbial growth, indicating that mechanosensory stimuli interact with chemosensory-guided decisions to select or avoid oviposition sites.

## 1. Introduction

Oviposition site selection is a critical factor in determining the survival rate of offspring in insect species. A nutritionally suitable resource may be heavily used by other insects and the offspring may suffer from intense competition. The females of *Drosophila suzukii* Matsumura (Diptera: Drosophilidae) have the ability to pierce the skin of ripening fruits and lay eggs into the flesh by using serrated ovipositors [1–3]. Because many other closely related *Drosophila* species lay eggs onto fermented fruits, this behaviour allows *D. suzukii* to use a carbohydrate-rich resource before interspecific competition becomes intense [4,5].

The behavioural shift to deposit eggs into ripening fruits must have been accompanied by changes not only in the ovipositor

morphology but also in the sensory systems used to evaluate the oviposition substrate. Karageorgi *et al.* [6] showed that when given the choice between ripe and rotten strawberry fruits, *D. suzukii* strongly preferred ripe over rotten fruit, whereas *Drosophila melanogaster* showed an opposite tendency and preferred rotten fruit, consistent with other studies [7,8]. In the same experiment, *Drosophila biarmipes*, a closely related species of *D. suzukii* showed no preference between ripe and rotten fruit, indicating that they are at an intermediate evolutionary stage between *D. suzukii* and *D. melanogaster*. It has also been shown in the same study that while *D. biarmipes* and *D. melanogaster* show similarly strong preferences for soft substrates, *D. suzukii* lay eggs onto both hard and soft agarose gel substrates, a pattern similar to other studies [4,9]. Therefore, these studies indicate that *D. suzukii* has widened the range of potential substrates to include those with different degrees of hardness and does not necessarily prefer a harder fruit surface [10–12]. Thus, hardness alone does not account for the strong preference for ripe fruits as an oviposition substrate. Other sensory modifications are also likely to underlie the radical shift to an unexploited resource in *D. suzukii* after divergence from the *D. biarmipes* lineage.

The evolutionary changes in the *D. suzukii* chemosensory system and response to attractants from ripening fruits have been documented [6,13,14], but possible repellents of fermenting fruits have not been investigated in detail. As shown in the previous studies, inoculation of the substrate from *D. melanogaster* adults significantly reduced the number of eggs laid by *D. suzukii* [15,16]. The factors causing such aversive behaviour are not known. The deposition of aggregation pheromones is one likely factor [17–19]. In addition, microbial populations on fermenting fruits originating from the surrounding environment as well as individuals that have visited the fruit represent another source of aggregation signals. The presence of non-pathogenic microbes guides a wide array of behavioural decisions in insects, including adult aggregation, feeding decisions and oviposition choice [20–24]. Partnering with commensal microbes provides several benefits for insect hosts including protection from pathogenic microbes, increased access to nutritional resources and improved offspring survival [25]. The response of *D. suzukii* oviposition to the microbial environment has been largely unstudied but may represent an essential aspect of the new host exploitation in this species.

Assessing the fruit condition and making the decision to select the oviposition site involve an integration of multiple sensory cues. It has been shown that *D. suzukii* has the ability to make complex decisions between healthy and fermenting fruits depending on the availability of the resource [8]. We hypothesize that one of the unexplored factors that *D. suzukii* sense could be the commensal microbes on fermenting fruits that have been deposited by conspecific and heterospecific individuals. The avoidance of fruits with chemicals (metabolites) from such microbes may be an effective strategy to access the fruit resource before overripening or fermenting reactions proceed. However, other information such as texture also is likely to be perceived and used to make ultimate decisions. Indeed in *D. melanogaster*, mechanosensory (texture) and chemosensory (taste) information are integrated to direct feeding and oviposition decisions [26–28]. Therefore, it is an intriguing question as to how different sensory information is processed and integrated in *D. suzukii* in comparison with *D. biarmipes* and *D. melanogaster*, both of which have different decision-making criteria for choosing oviposition sites.

In this study, we investigate the effects of commensal microbes on oviposition site preferences, both independent of and in combination with the effect of the substrate hardness, in *D. suzukii*, *D. biarmipes* and *D. melanogaster*. In our assay, *D. suzukii* exhibited a strong avoidance of microbes transferred from other flies. This response was distinct from the other two species, suggesting that the behaviour has evolved in the lineage leading to *D. suzukii* after the split from *D. biarmipes*. Furthermore, we tested the combinatorial effect of the hardness and the presence or absence of microbes on the oviposition site selection. The mechanical stimuli provided by substrate hardness superseded the influence of microbial chemical signals. We show that this property was conserved among the three species despite differential preference towards hardness and microbial stimuli.

## 2. Material and methods

### 2.1. Fly strains

The following strains were used to compare the oviposition site preference: *D. suzukii* strain Hilo collected in Hilo, Island of Hawai'i, USA in 2017, *D. biarmipes* strain MYS118, collected in Mysore, India, in 1981, and

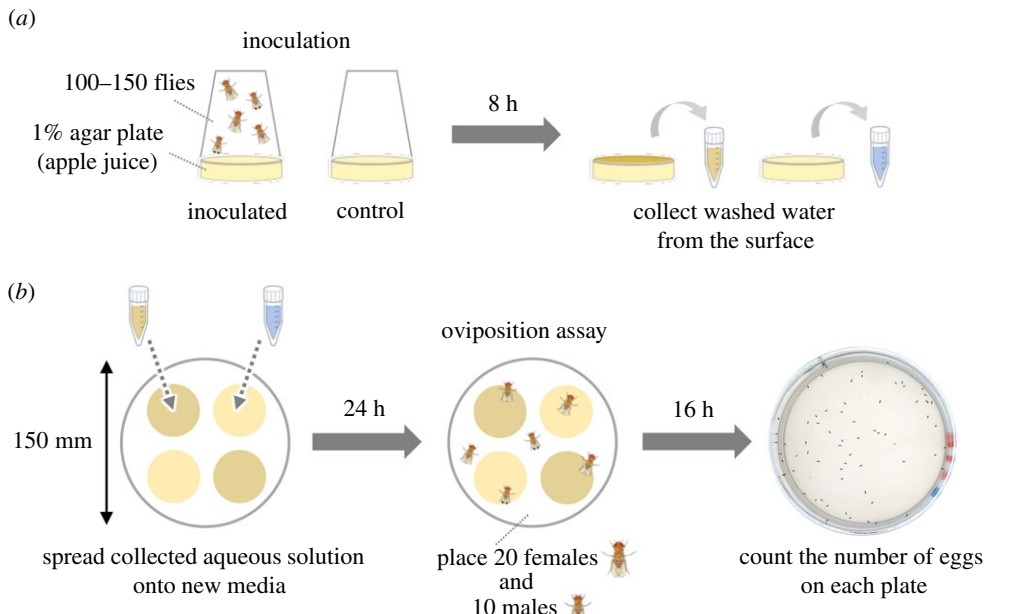

**Figure 1.** Experimental scheme of the oviposition assay to quantify response to microbes deposited by flies on the surface of media. (a) Washed water collected from the surface of inoculated and control plates. (b) Oviposition assay using media inoculated with solutions from (a) for 24 h.

*D. melanogaster* strain Canton S BL#9515. All the strains were maintained at 25 ± 1°C under the 12 h light: 12 h dark light cycle. All flies were fed with standard corn meal food mixed with yeast, glucose and agar.

## 2.2. Oviposition assay to test the preference for substrates with microbial growth

The procedure is illustrated in figure 1. Inoculation was conducted by using *D. melanogaster* (3–7 days after eclosion), *D. biarmipes* (3–7 days after eclosion) or *D. suzukii* (7–14 days after eclosion). One hundred to 150 flies were placed into the inoculation chamber without anaesthesia and left for 8 h. An inoculation chamber consists of a plastic cup (100 ml, Tri-Corner Beakers) and a petri dish (57 mm diameter × 16 mm height, IWAKI 1010-060) filled with 5 ml 1% agar (*Drosophila* agar type II, Apex) in apple juice (SUNPACK, JAN code: 4571247510950) diluted to 50%. No flies were placed into the control inoculation chamber. After inoculation, the surface of the substrate was washed with 1 ml distilled water by pipetting 10 times. Wash solutions (100 µl) from inoculated or control plates (figure 1a) were spread onto a new agar plate (40 mm diameter × 13 mm height, Azunol 1-8549-01) and incubated for 24 h at 25 ± 1°C. Microbial colonies were visible on the media spread with aqueous solution from the inoculated media after 24 h incubation.

The oviposition assay was conducted with a petri dish chamber (150 mm diameter × 20 mm height, IWAKI 3030-150) containing four 40 mm diameter petri dishes with two types of media placed alternatively (figure 1b). Twenty females and 10 males were placed into the chamber without anaesthesia within 3 h before the dark cycle and kept for 16 h in the dark condition. The assay was conducted under the condition of 25 ± 1°C and 50 ± 5% relative humidity. The photo image of each petri dish with substrate was taken by a camera (Olympus DP73) with transmitted light from the bottom. The number of eggs on each substrate was counted.

The preference index (PI) for the substrate with microbial growth was calculated by using the following formula:

$$\text{preference index (PI) for substrate with microbial growth} = \frac{N_{\text{inoculated}} - N_{\text{control}}}{N_{\text{inoculated}} + N_{\text{control}}},$$

where $N_{\text{inoculated}}$ and $N_{\text{control}}$ are the total numbers of eggs on the substrates with microbial growth and the control plates, respectively.

To confirm that the PI measurements for substrates inoculated with microbial colonies reflect the activity of microbes, collected solutions from the inoculated media were filter sterilized using a syringe filter (0.22 µm Millex®-GV Filter Unit). After washing the surface of the inoculated medium

by repeatedly pipetting 1.2 ml distilled water 10 times, the aqueous solution was filtered and used in the oviposition assay as described earlier.

## 2.3. Oviposition preference assay for substrate hardness, with and without microbes

Inoculant from *D. melanogaster* was collected from three inoculation chambers, pooled and divided into 24 (8 × 3 species) 40 mm diameter petri dishes with medium. Plates without any solution were used for the assays that did not test microbial inoculation. The remaining steps were the same as in §2.2. The PI for the soft substrate was calculated by using the following formula:

$$\text{preference index (PI) for soft substrate} = \frac{N_{1\% \text{ agar}} - N_{3\% \text{ agar}}}{N_{1\% \text{ agar}} + N_{3\% \text{ agar}}},$$

where $N_{1\% \text{ agar}}$ and $N_{3\% \text{ agar}}$ are the total numbers of eggs on the 1% and 3% agar media, respectively.

## 2.4. 16S-rRNA gene sequencing of microbial colonies used for the oviposition assays

To collect the microbes tested for the oviposition assays, the surface of the inoculated substrate was washed with distilled water as described earlier. The solution was diluted to 200 µl total volume and spread onto a petri dish (90 mm diameter × 16 mm height, IWAKI SH90-15) filled with 10 ml apple juice agar as described earlier. The media were incubated for 24 to 40 h at 25 ± 1°C, and single colonies were selected randomly for DNA extraction. Each colony was picked with a 10 µl pipette tip, suspended in 20 µl of sterile water and incubated for 15 min at 95°C after adding 20 µl 100 mM NaOH. Then, 4.4 µl of 1 M Tris–HCl pH 7.0 was added to each sample and used as template DNA.

Colony PCR was performed with 16S-rRNA universal primers 8F (AGAGTTTGATCMTGGCTCAG) [29,30] and 1492R (GGYTACCTTGTTACGACTT) [31,32] in a 30 µl reaction using Ex Taq (TaKaRa). Amplification condition for the PCR included an initial denaturation step of 95°C for 3 min, followed by 35 cycles of 95°C for 30 s, 53 or 55°C for 30 s and 72°C for 60 s, and a final extension step of 72°C for 5 min. Reaction products were checked for size and purity on 1% agarose gel and were sequenced after purification by using either BrilliantDye Terminator Cycle Sequencing Kit v. 2.1 (Nimagen) and a 3130 xl DNA Analyzer (Thermo Fisher Science) or BigDye Terminator v. 3.1 Cycle Sequencing Kit (Thermo Fisher Science) and a 3170xl DNA Analyzer (Thermo Fisher Science). Sequences were aligned by using MEGA7 [33] and trimmed from the nucleotide positions 61 to 628 of the *Escherichia coli* reference sequence (CP023349.1:226,883-228,438). The genus level identity of each sequence was assigned by the highest score entries in the NCBI database, '16S ribosomal RNA (Bacteria and Archaea type strains)' (as of 28 May 2020) by local BLAST (BLAST + 2.10.0).

# 3. Results

The oviposition site preference of *D. suzukii* for ripening fruits relies on shifts in mechanosensation as well as chemosensation [6]. Recent work has shown that consistent with their preference towards ripening fruits over fermenting fruits, *D. suzukii* females tend to lay more eggs on non-inoculated media compared with media inoculated by *D. melanogaster* [15]. Our study focused on determining whether microbial presence and the hardness of the oviposition substrate form the basis of *D. suzukii* oviposition decisions.

## 3.1. Oviposition site preference against the presence of microbes

Oviposition can be influenced by pheromones or microbial presence. To distinguish between these two possibilities, we first established a method to test only the contribution of microbial growth to oviposition site preference. A water wash was used to collect substances deposited by adult flies, and the inoculum was applied to sterile media (figure 1a). Many of the known pheromones used for *Drosophila* chemical communication are hydrophobic hydrocarbons, wax esters and wax alcohols [34] and are thus not soluble in water and unlikely to be transferred in the water wash. After incubation, microbial colonies were visible on the inoculated media. Media that had been exposed to water wash from control chambers did not have visible colonies.

The results from the oviposition assay on soft medium (1% agar) indicated that *D. suzukii* avoided oviposition substrates with microbial colonies (figure 2a, electronic supplementary material, table S1).

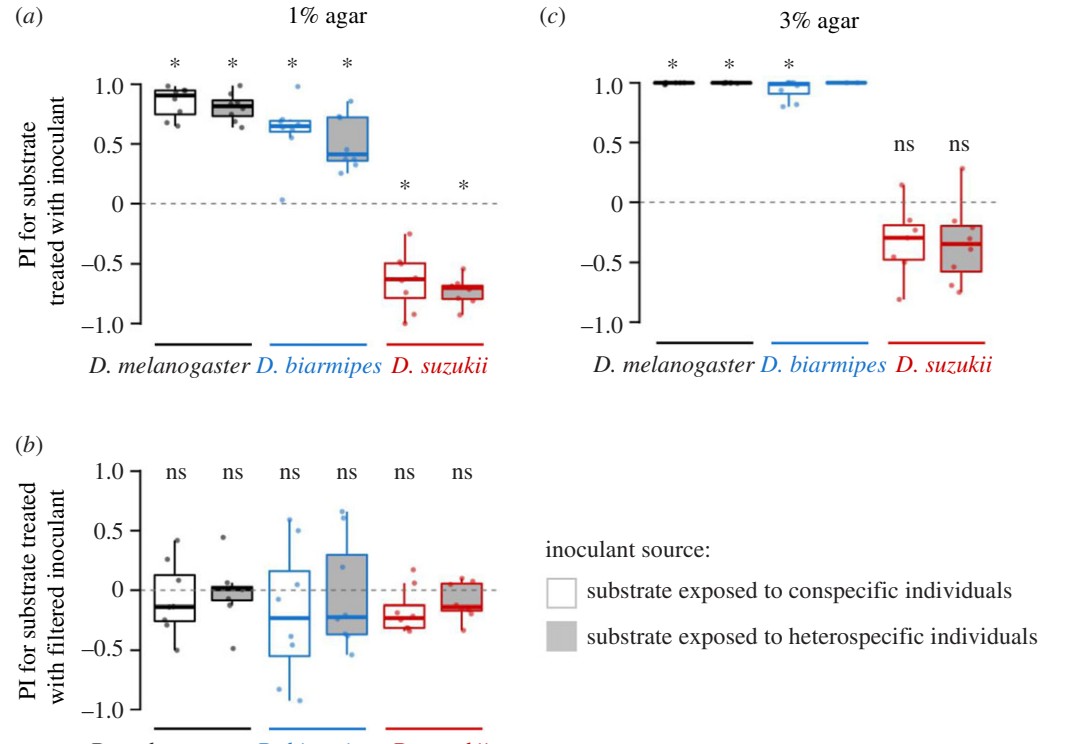

**Figure 2.** Comparisons of the preference indices (PIs) of *D. melanogaster*, *D. biarmipes* and *D. suzukii* for oviposition substrates treated with inoculant from conspecific (open boxplots) or heterospecific (filled boxplots in grey) flies. (*a*) The PIs assayed on soft substrate (1% agar medium) with and without inoculant treatment (microbial growth). (*b*) The PIs assayed on 1% agar medium for substrates treated with sterile filtered solutions of inoculant. (*c*) The PIs assayed on hard oviposition substrate (3% agar medium) with and without inoculant treatment (microbial growth). Control substrates were treated with solutions from non-exposed (non-inoculated) substrate in all assays. Species used for heterospecific inoculations were conducted using *D. suzukii* for *D. melanogaster* assay, and *D. melanogaster* for *D. biarmipes* and *D. suzukii* assays. Results from assays with fewer than 10 eggs on either substrate were excluded from the analysis. Box signifies the upper and lower quartiles and horizontal bar indicates median. Upper and lower whiskers represent maximum and minimum 1.5 × interquartile range, respectively. The difference from PI = 0 (no preferences) was tested by Wilcoxon signed-rank test with Bonferroni correction for multiple comparisons (six tests). $^*p < 0.05$, ns $p \geq 0.05$.

Given a choice between substrates with aqueous solutions from inoculated and non-inoculated media, the *D. suzukii* PI was significantly less than 0, indicating that the microbial growth discouraged oviposition. By contrast, *D. melanogaster* preferred ovipositing on substrates with the microbial growth (figure 2*a*), indicating that the presence of microbes positively influenced the choice of oviposition site for this species. To trace the evolutionary trajectory of this preference, we also conducted the same experiments using *D. biarmipes*, a closely related species to *D. suzukii*. Remarkably, as with *D. melanogaster*, the microbes positively influenced the oviposition site choice of *D. biarmipes* (figure 2*a*), indicating that the preference for ovipositing at sites with commensal microbes is the ancestral state among these species and that *D. biarmipes* still retain this characteristic. These results were consistent when using microbes from conspecific and heterospecific inoculation (figure 2*a*). Thus, the drastic change from the tendency to lay more eggs to fewer eggs on the substrate with microbial growth is predicted to have occurred in the lineage leading to *D. suzukii* after the separation from the *D. biarmipes* lineage.

To confirm that the presence of microbes in the water wash is the primary factor in guiding oviposition, we passed the collected aqueous solution through a 0.22 μm filter to remove microbes and large food particles while keeping nutrients, metabolites and other small molecules found in faeces. In all species, filter sterilization of the inoculant eliminated both positive and negative oviposition preferences (figure 2*b*, electronic supplementary material, table S2). Therefore, microbes that can be removed by a 0.22 μm filter are likely to be the main factor affecting oviposition site preferences.

To identify the main bacterial species that were present in the water washes of inoculated media, we sampled microbial colonies from the medium after 24–40 h of growth and performed PCR amplification of the 16S-rRNA gene sequence. The bacterial species classified at the genus level and the frequencies estimated from the sampled colonies are shown in electronic supplementary material, figure S1 and

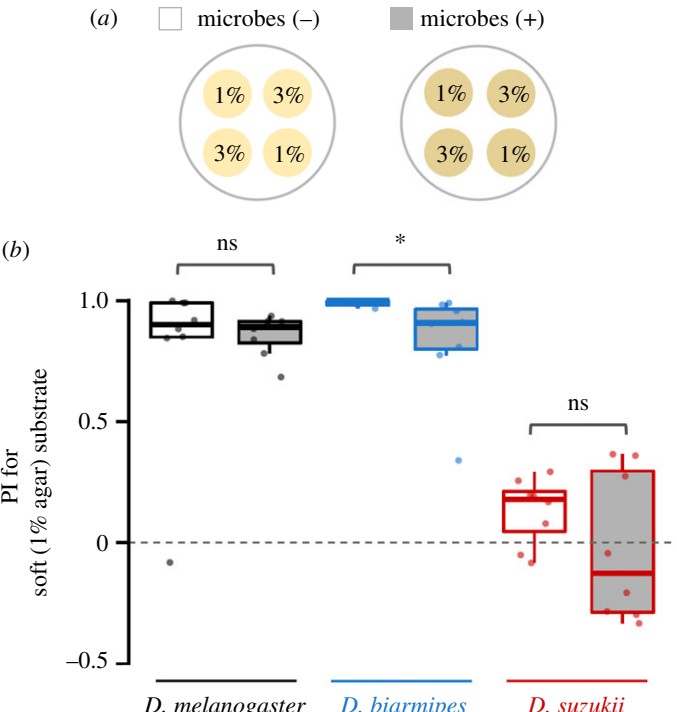

**Figure 3.** Preference indices (PIs) for the soft substrate with and without microbes. (a) The substrate placement in the chambers for the oviposition assay. '1%' and '3%' indicate soft (1% agar medium) and hard (3% agar medium) oviposition substrates, respectively. The microbe (+) chambers have been treated with inoculant collected from substrate surface exposed to *D. melanogaster*; microbial (−) chambers were treated with inoculant from non-exposed surfaces. (b) The preference indices (PIs) for soft oviposition substrate in the absence (open boxplots) and presence (filled boxplots in grey) of microbes. Results from assays with fewer than 10 eggs on either substrate were excluded from the analysis. Box signifies the upper and lower quartiles and horizontal bar indicates median. Upper and lower whiskers represent maximum and minimum $1.5 \times$ interquartile range, respectively. Statistical significance was tested by permutation test with Bonferroni correction for multiple comparisons (six tests). $^{*}p < 0.05$, ns $p \geq 0.05$.

electronic supplementary material, tables S3–S5. The bacteria used for our oviposition preference assay were mostly from the *Acetobacter* and *Gluconobacter* genera.

## 3.2. Combinatorial effect of the presence of microbes and the hardness of the oviposition substrate

In addition to chemosensory signals, another factor that is known to affect *Drosophila* oviposition site preference is the hardness of the substrate. Choice assays using agarose media with different degrees of hardness have shown that *D. suzukii* females exhibit a much weaker preference towards softer substrates, resembling damaged and fermented fruits, compared with *D. biarmipes* and *D. melanogaster* [6]. To investigate the combinatorial effect of hardness and microbial growth, we conducted choice assays using hard oviposition substrate (3% agar medium) with and without the presence of microbes (figure 2c, electronic supplementary material, table S6).

When substrates were hard, *D. melanogaster* and *D. biarmipes* showed a PI close to 1, which is indicative of even stronger preferences for ovipositing on media with microbial growth than when using 1% agar media (figure 2a). Interestingly, the aversion to substrates with microbial growth exhibited by *D. suzukii* was reduced when the harder 3% media were used. No significant preference or aversion was detected for microbial growth when the substrates in the oviposition chamber were all hard (figure 2c). From the outcome of this combinatorial assay, it was clear that the hardness of the substrate modifies the preferences against microbes.

Next, we investigated whether the choice between soft (1%) and hard (3%) agar media was affected by the presence of microbes (figure 3a). Our experimental results using 1% and 3% agar media without microbes showed that *D. suzukii* had no strong preference towards either substrate, in contrast to the strong preference exhibited by the other two species (figure 3b, electronic supplementary material, table S7),

which was consistent with the previous study [4,6,9]. Interestingly, whether the microbes were present or not did not affect the PI between soft and hard substrates in *D. melanogaster* and *D. suzukii*. The preference towards the softer substrate became significantly weaker when microbes were present than when they were absent in *D. biarmipes*, but only slightly. These results indicate that rather than the presence or absence of microbial growth, the hardness of the substrate is the dominant factor determining the oviposition site selection in *D. melanogaster* and *D. biarmipes*. In *D. suzukii*, the lack of preference to lay more eggs on either the soft or hard substrate was persistent and unaltered by the microbial growth.

# 4. Discussion

## 4.1. Commensal microbes deposited by flies affect oviposition site preferences in *D. suzukii*, *D. biarmipes* and *D. melanogaster*, and the preference of *D. suzukii* is distinct from that of the other species

Fruit flies like many other insects coexist with a community of gut microbes, the composition of which can vary to a large extent due to various field and laboratory conditions [35–38]. To elucidate whether chemicals emitted from gut microbes function as intraspecific or interspecific behavioural cues, we examined the influence of fly-deposited microbes on oviposition behaviour.

Our results show that egg-laying decisions in *Drosophila* are strongly influenced by the presence of microbial growth, suggesting that they are sensitive to microbe-derived cues. When given a choice using soft media, *D. suzukii* avoided media inoculated with commensal microbes, in contrast to *D. melanogaster* and *D. biarmipes*, both of which showed strong oviposition preferences towards microbe-rich media (figure 2). The significant change in oviposition site preference must have occurred in the *D. suzukii* lineage after the split from *D. biarmipes* consistent with the timing of the host shift to ripening fruits. Therefore, the change in microbial preference may have been associated with the new niche exploitation in this lineage.

## 4.2. Acetic acid bacteria differentially affect oviposition behaviour among *Drosophila* species

The bacterial species used for oviposition preference assays consisted mainly of *Acetobacter* and *Gluconobacter,* both members of the Acetobacteraceae family commonly found in the guts of laboratory-raised and wild fruit fly species [36] including *D. suzukii* [39,40]. These acetic acid bacteria provide benefits for host flies by accelerating growth and offering protection from pathogenic bacteria [41,42]. Some previous studies on wine grapes have indicated that *D. suzukii* is capable of vectoring acetic acid bacteria that contribute to the fermentation process of the fruits [43,44]. Nevertheless, the colonies grown on the media are not likely to represent the actual composition of fly-associated microbiota in the wild since growth is restricted by diet and the type of media used (agar in apple juice). Flies from natural populations exhibit a more diverse microbiome [37,45]. In addition, our characterization of the microbiome focused only on bacterial species. It is likely that yeast, which is a common symbiont for drosophilids [46], is also a part of the inoculum and contributes to oviposition preference [47].

*Drosophila melanogaster*, *D. biarmipes* and *D. suzukii* exhibited different proportions of *Acetobacter* and *Gluconobacter* (electronic supplementary material, figure S1 and electronic supplementary material, tables S3–S5). However, there were no differences in the responses of the three *Drosophila* species to conspecific or heterospecific inoculants, indicating that both *Acetobacter* and *Gluconobacter* have similar effects on the oviposition site choice (figure 2). While *D. suzukii* showed a clear aversion for ovipositing on inoculated soft media (figure 2*a*), the response of females to *Gluconobacter* volatiles may be context dependent. A previous study showed that females starved for 24 h exhibit clear attraction to *Gluconobacter* in an olfactometer bioassay [48]. Taken together with our observation that *D. suzukii* avoids egg laying in the presence of *Gluconobacter* colonies, it is clear that reproductive and feeding site preferences can be clearly decoupled in this species. Microbial cues that are attractive for feeding may be aversive for oviposition.

## 4.3. Chemical cues mediating the differential preference against microbes await further investigation

In studies searching for oviposition deterrents for the pest management of fruit crops, at least two chemicals, geosmin and octenol (1-octen-3-ol), both of which are components of volatile

metabolites from microorganisms present in rotting fruits, induced aversive responses in *D. suzukii* [49]. However, because these chemicals are known repellents in *D. melanogaster* as well [50,51], the aversion to these microbial compounds is not likely to underlie the *D. suzukii* specific shift in oviposition site.

A study using *D. melanogaster* indicated that female oviposition is guided by sucrose, a gustatory cue used to sense fermentation by lactic acid-producing *Enterococci* bacteria [52]. Interestingly, the olfactory system was shown to be dispensable for ovipositional attraction to these microbes. In contrast, the inhibition of synaptic transmission in sweet sensing gustatory neurons, *Gr5a* and *Gr64a* neurons, impaired the oviposition preference towards fermentation sources. Whether sucrose sensing also mediates the avoidance of acetic acid bacteria in *D. suzukii* would be an intriguing question to pursue. Nevertheless, Silva-Soares *et al.* [4] showed that *D. suzukii* and *D. biarmipes* have similar oviposition preferences towards sites with a low protein (yeast) to carbohydrate (sucrose) ratio, suggesting that a differential response to sucrose is not likely to explain the contrasting response to acetic acid bacteria products. The volatiles emanating from microorganisms may also be playing a substantial role in making decisions. Thus, the microbe-derived chemical cues that govern oviposition response await further investigation.

One feature of the experimental design that may impact oviposition decisions is the ventilation of the behavioural chamber. The arena housing the oviposition chambers of our study was not ventilated. The lack of ventilation may obscure the choice within a chamber due to a buildup of odorants or bias the preferences because of an unnaturally concentrated cue. However, neither effect appeared to be a substantial factor in our experiments since there were both instances where a clear choice or no choice was made (figure 2*a*,*c* and figure 3*b*). It also appeared to be not a critical factor in a previous study investigating the effects of acids on positional responses and oviposition preferences using *D. melanogaster* [53]. Nevertheless, it should be noted that we cannot totally exclude the possibility that the lack of ventilation may have caused some subtle biases in preference.

## 4.4. Oviposition site hardness supersedes the *D. suzukii* aversion to microbial presence

Integration of different types of stimuli is essential for critical decision-making processes such as the selection of egg deposition sites, a choice that has large influences on the early life performances of the offspring. In *D. melanogaster*, neural circuits governing oviposition site combine information from different modalities [53,54]. Recently, several studies [27,28] elucidated an underlying molecular mechanism for integrating mechanosensory and chemosensory information to make egg-laying decisions in *D. melanogaster*. Our results reveal that two different classes of sensory cues, substrate hardness and the presence of microbes, are integrated in *D. suzukii* oviposition decisions in a manner that is distinct from *D. biarmipes* and *D. melanogaster* (figures 2 and 3). The avoidance of microbes displayed by *D. suzukii* was evident only in the context of a soft substrate (figure 2*a*) but not a hard one (figure 2*c*). These results suggest that mechanical cues from surface hardness take precedence over decisions guided by microbial cues. By contrast, the preference exhibited by both *D. melanogaster* and *D. biarmipes* towards microbe-inoculated surfaces strengthened when hard substrates were used (figure 2*c*), indicating a similar integration of mechanical and microbial chemical cues. Conversely, microbial presence did not affect the choice between hard and soft substrates in all the three species (figure 3).

These results indicate that mechanical and chemical stimuli are not processed additively in these species. The surface hardness modifies the response to microbial cues but not vice versa. Interestingly, previous studies showed that in female *D. melanogaster*, the presence of chemicals, sucrose and/or fruit juice ingredient obviates the preference for ovipositing on softer surfaces [27,28]. The discrepancy between the direction of interference between mechanical and chemical stimuli suggests that the hierarchy of cues used in oviposition may depend on the nature of the chemical stimulus.

## 4.5. The integration of mechanical cues and microbial stimuli is conserved in oviposition choice and reflect differences in ecology

The hardness of the substrate assayed using 1% and 3% agar media is intended to mimic damaged fermenting fruits and intact ripening fruits, respectively. However, the agar media used in our assay have uniform texture. This feature may not completely reflect the condition of the real ripening fruits with partially damaged skin in the field. Indeed, it has been shown that egg-laying decisions of flies

depend on whether the fruit is injured or not [2,8]. Nevertheless, despite this caveat, our findings in this study can still be interpreted conceptually in the context of ecology of *D. suzukii*.

In the early fruiting season when all the fruits are hard or likely to have only a small amount of commensal microbial cues left by other flies, *D. suzukii* females may lay eggs onto any available fruits. This scenario is consistent with the results of our assays using only hard substrate (figure 2*c*) or only non-inoculated substrates (figure 3). During the ripening period, when fruits become softer and partially damaged, the females may choose fruits with less abundant fermentation cues presumably to avoid competition with other species. This prediction is consistent with our results using only soft substrate (figure 2*a*). In late fruiting season when the majority of the fruits are on the ground, damaged and rotten, the females may readily lay eggs onto suboptimal fermenting fruits, the situation resembling our assays using only inoculated substrates (figure 3). These explanations are consistent with the study by Kienzle *et al.* [8], which showed that *D. suzukii* exhibit stronger preferences towards ovipositing in healthy fruits when healthy and fermenting fruits are both abundant compared with when the former are less abundant. The context-dependent optimization through seasonal change in host fruit condition might explain the evolutionary background of our findings where substrate hardness takes precedence over microbial presence in the decision to oviposit in this species.

Although surface hardness interacts with the response to commensal microbe cues in *D. biarmipes* and *D. melanogaster* as in *D. suzukii*, there may be some qualitative differences in ecological context between these species. *Drosophila biarmipes* and *D. melanogaster* show a strong preference towards soft substrates inoculated with microbes, and their preferences for microbes are enhanced when the substrate is hard (figure 2). In the field, it may be the case that flies are more likely to use hard fruits in the presence of a microbial signature, which may be indicative of an immediate onset of fermentation when the skins become partially damaged as the fruits ripen. In contrast to *D. suzu*kii, both *D. biarmipes* and *D. melanogaster* tend to prefer soft substrates even when all the substrates in the vicinity have microbial growth (figure 3), indicating that mechanical cues supersede microbial presence in oviposition site selection. Therefore, *D. suzukii* may have rapidly evolved to adjust the manner in which mechanical and chemical stimuli are integrated to maximize the offspring performance by an egg-laying strategy that is different from other closely related species.

Data accessibility. The datasets supporting this article have been uploaded as part of the electronic supplementary material.

Authors' contributions. A.S., J.Y.Y. and A.T. conceived the research and designed the experiments. A.S. performed the experiments. A.S. and K.M.T. analysed the data. A.S. and A.T. drafted the manuscript. All authors gave final approval for publication.

Competing interests. We declare we have no competing interests.

Funding. This work was supported by JSPS KAKENHI (grant no. JP19H03276) awarded to A.T., Department of Defense United States Army Research Office (grant no. W911NF1610216) and the National Institutes of Health (grant no. 1P20GM125508) awarded to J.Y.Y.

Acknowledgments. We are grateful to K. Nakayama and N. Yoneishi for excellent technical assistance, the UHM Microbial Genetics and Analytical Laboratory for use of facilities, Eurofins Genomics K.K. for Sanger sequencing, and members of the Takahashi lab and Yew lab for helpful discussions.

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
