## [Reviewer comments · Royal Society Open Science]

Review History

RSOS-201601.R0 (Original submission)

Review form: Reviewer 1 (Marko Rohlf)

Is the manuscript scientifically sound in its present form?

Yes

Are the interpretations and conclusions justified by the results?

No

Is the language acceptable?

Yes

Do you have any ethical concerns with this paper?

No

Have you any concerns about statistical analyses in this paper?

No

Recommendation?

Major revision is needed (please make suggestions in comments)

Comments to the Author(s)

I think this article makes an interesting contribution to the ecology of *Drosophila suzukii*. I have the following points of criticism which should be addressed to give the article the necessary attention by readers. I have made specific remarks as comments directly in the pdf document, so my comments here are only general:

1. it would help for the overall understanding if the use of terms and procedures were better justified. In particular, one can only guess why 'substrate hardness' is so important in this article and what the methodological approach actually investigates. In my view, a general research question is also missing.
2. several conclusions are in my opinion not allowed or too strong (see my comments in the pdf document, Appendix A). And here also the problem arises that it is not clear why substrate hardness should be so central, this will only be clarified in the discussion. Personally I also think that this discussion of substrate hardness is only incomplete. I miss the point in the discussion that *D. melanogaster* and others need injured fruits, they can also be unripe and of high substrate hardness, ok, then they might not be so super attractive but accessible. But ripe and internally soft fruits are irrelevant for *D. melanogaster*, because there is no wounded site; and *D. suzukii* cannot perceive the soft interior of a ripe fruit by only having access to the fruit skin. Indeed, egg laying decisions depend on whether the fruit is injured or not, as we have shown in our paper, Kienzle et al. 2020. The discussion would benefit if this aspect of the fruit injury, which results in a 'soft' substrate, were to be compared to the manipulated substrate hardness with agar.

Review form: Reviewer 2

Is the manuscript scientifically sound in its present form?

Yes

Are the interpretations and conclusions justified by the results?

Yes

Is the language acceptable?

Yes

Do you have any ethical concerns with this paper?

No

Have you any concerns about statistical analyses in this paper?

Yes

Recommendation?

Accept with minor revision (please list in comments)

Comments to the Author(s)

This is a study that contributes to a better understanding of the biology of several *Drosophilid* fly species, together with the role of fruit firmness and presence of micro-organisms.

The authors did a good job to conduct rigorous science and clearly described their results. The publication should be accepted with minor revisions.

The authors are encouraged to describe why the oviposition arenas were not ventilated, and what the shortcomings of this experiment were. With ventilation, a volatile cloud is removed and will

likely contribute to more focused selection of certain sites that either contain, or do not contain the studied microorganisms.

The authors should cite work (2 publications of Ioriatti et al., there are several others that are also somewhat related on winegrape, which describe the interactions of Drosophilids as vectors of several of the microorganisms mentioned in this paper, it will strengthen their arguments and evidence of scholarship. Although this is not the focus of the current paper, additional discussion of volatiles emanating from the microorganisms may be important as well.

Decision letter (RSOS-201601.R0)

Dear Dr Takahashi

The Editors assigned to your paper RSOS-201601 " *Drosophila suzukii* avoidance of microbes in oviposition choice" have now received comments from reviewers and would like you to revise the paper in accordance with the reviewer comments and any comments from the Editors. Please note this decision does not guarantee eventual acceptance.

Please submit your revised manuscript and required files (see below) no later than 21 days from today's (ie 23-Nov-2020) date. Note: the ScholarOne system will 'lock' if submission of the revision is attempted 21 or more days after the deadline. If you do not think you will be able to meet this deadline please contact the editorial office immediately.

Kind regards,
Andrew Dunn
Royal Society Open Science Editorial Office
Royal Society Open Science

on behalf of Professor Simon Sprecher (Associate Editor) and Kevin Padian (Subject Editor)
openscience@royalsociety.org

Editor comments:

Thank you for your submission. Should you need more time to make revisions, please contact our editorial office. Best wishes.

Associate Editor Comments to Author (Professor Simon Sprecher):

Associate Editor: 1

Comments to the Author:

The reviewers are overall positive about the manuscript, but raise a few points that have to be addressed.

Reviewer comments to Author:

Reviewer: 1

Comments to the Author(s)

I think this article makes an interesting contribution to the ecology of *Drosophila suzukii*. I have the following points of criticism which should be addressed to give the article the necessary attention by readers. I have made specific remarks as comments directly in the pdf document, so my comments here are only general:

1. it would help for the overall understanding if the use of terms and procedures were better justified. In particular, one can only guess why 'substrate hardness' is so important in this article and what the methodological approach actually investigates. In my view, a general research question is also missing.
2. several conclusions are in my opinion not allowed or too strong (see my comments in the pdf document). And here also the problem arises that it is not clear why substrate hardness should be so central, this will only be clarified in the discussion. Personally I also think that this discussion of substrate hardness is only incomplete. I miss the point in the discussion that *D. melanogaster* and others need injured fruits, they can also be unripe and of high substrate hardness, ok, then they might not be so super attractive but accessible. But ripe and internally soft fruits are irrelevant for *D. melanogaster*, because there is no wounded site; and *D. suzukii* cannot perceive the soft interior of a ripe fruit by only having access to the fruit skin. Indeed, egg laying decisions depend on whether the fruit is injured or not, as we have shown in our paper, Kienzle et al. 2020. The discussion would benefit if this aspect of the fruit injury, which results in a 'soft' substrate, were to be compared to the manipulated substrate hardness with agar.

Reviewer: 2

Comments to the Author(s)

This is a study that contributes to a better understanding of the biology of several *Drosophilid* fly species, together with the role of fruit firmness and presence of micro-organisms.

The authors did a good job to conduct rigorous science and clearly described their results. The publication should be accepted with minor revisions.

The authors are encouraged to describe why the oviposition arenas were not ventilated, and what the shortcomings of this experiment were. With ventilation, a volatile cloud is removed and will

likely contribute to more focused selection of certain sites that either contain, or do not contain the studied microorganisms.

The authors should cite work (2 publications of Ioriatti et al., there are several others that are also somewhat related on winegrape, which describe the interactions of Drosophilids as vectors of several of the microorganisms mentioned in this paper, it will strengthen their arguments and evidence of scholarship. Although this is not the focus of the current paper, additional discussion of volatiles emanating from the microorganisms may be important as well.

===PREPARING YOUR MANUSCRIPT===

===PREPARING YOUR REVISION IN SCHOLARONE===

Please ensure that you include a summary of your paper at Step 2 'Type, Title, & Abstract'. This should be no more than 100 words to explain to a non-scientific audience the key findings of your

research. This will be included in a weekly highlights email circulated by the Royal Society press office to national UK, international, and scientific news outlets to promote your work.

Author's Response to Decision Letter for (RSOS-201601.R0)

See Appendix B.

Decision letter (RSOS-201601.R1)

This year has been very difficult for everyone, and we want to take the opportunity to thank you for your continued support in 2020.

The Royal Society Open Science editorial office will be closed from the evening of Friday 18 December 2020 until Monday 4 January 2021. We will not be responding during this time. If you have received a deadline within this time period, please contact us as soon as possible to allow us to extend the deadline. If you receive any automated messages during this time asking you to meet a deadline, we offer apologies and invite you to respond after the festive period or during normal working hours.

With our best for a peaceful festive period and New Year, and we look forward to working with you in 2021.

Dear Dr Takahashi,

It is a pleasure to accept your manuscript entitled "*Drosophila suzukii* avoidance of microbes in oviposition choice" in its current form for publication in Royal Society Open Science.

You can expect to receive a proof of your article in the near future. Please contact the editorial office (openscience@royalsociety.org) and the production office (openscience_proofs@royalsociety.org) to let us know if you are likely to be away from e-mail contact – if you are going to be away, please nominate a co-author (if available) to manage the proofing process, and ensure they are copied into your email to the journal.

on behalf of Professor Simon Sprecher (Associate Editor) and Kevin Padian (Subject Editor)
openscience@royalsociety.org

Follow Royal Society Publishing on Twitter: @RSocPublishing
Follow Royal Society Publishing on Facebook:
<https://www.facebook.com/RoyalSocietyPublishing.FanPage/>

Read Royal Society Publishing's blog:
<https://royalsociety.org/blog/blogsearchpage/?category=Publishing>

Appendix A**ROYAL SOCIETY
OPEN SCIENCE*****Drosophila suzukii* avoidance of microbes in oviposition
choice**

Journal:	Royal Society Open Science
Manuscript ID	RSOS-201601
Article Type:	Research
Date Submitted by the Author:	05-Sep-2020
Complete List of Authors:	Sato, Airi; Tokyo Metropolitan University, Department of Biological Sciences Tanaka, Kentaro; Tokyo Metropolitan University, Department of Biological Sciences Yew, Joanne; University of Hawai'i at Manoa, Pacific Biosciences Research Center Takahashi, Aya; Tokyo Metropolitan University, Department of Biological Sciences; Tokyo Metropolitan University, Research Center for Genomics and Bioinformatics
Subject:	ecology < BIOLOGY, evolution < BIOLOGY, microbiology < BIOLOGY
Keywords:	mechanosensory stimulus, decision-making, acetic acid bacteria, Gluconobacter , Acetobacter , spotted-wing drosophila
Subject Category:	Organismal and Evolutionary Biology

Author-supplied statements

Relevant information will appear here if provided.

Ethics

Does your article include research that required ethical approval or permits?:

This article does not present research with ethical considerations

Statement (if applicable):

CUST_IF_YES_ETHICS :No data available.

Data

It is a condition of publication that data, code and materials supporting your paper are made publicly available. Does your paper present new data?:

Yes

Statement (if applicable):

The datasets supporting this article have been uploaded as part of the Supplementary Material.

Conflict of interest

I/We declare we have no competing interests

Statement (if applicable):

CUST_STATE_CONFLICT :No data available.

Authors' contributions

This paper has multiple authors and our individual contributions were as below

Statement (if applicable):

A.S., J.Y.Y., and A.T. conceived the research and designed the experiments. A.S. performed the experiments. A.S. and K.M.T. analysed the data. A.S. and A.T. drafted the manuscript. All authors gave final approval for publication.

Drosophila suzukii* avoidance of microbes in oviposition choice**Airi Sato¹, Kentaro M. Tanaka¹, Joanne Y. Yew², Aya Takahashi^{1,3}¹Department of Biological Sciences, Tokyo Metropolitan University, 1-1 Minamiosawa, Hachioji 192-0397, Japan²Pacific Biosciences Research Center, University of Hawai'i at Mānoa, 1993 East West Road, Honolulu, HI 96822, United States³Research Center for Genomics and Bioinformatics, Tokyo Metropolitan University, 1-1 Minamiosawa, Hachioji 192-0397, Japan**Keywords:** mechanosensory stimulus, decision-making, acetic acid bacteria, *Gluconobacter*, *Acetobacter*, spotted-wing *DrosophilaAbstract**

While the majority of *Drosophila* species lay eggs onto fermented fruits, females of *D. suzukii* pierce the skin and lay eggs into ripening fruits using their serrated ovipositors. The changes of oviposition site preference must have accompanied this niche exploitation. In this study, we established an oviposition assay to investigate the effects of commensal microbes deposited by conspecific and heterospecific individuals, and showed that presence of microbes on the oviposition substrate enhances egg-laying of *D. melanogaster* and *D. biarmipes*, but discourages that of *D. suzukii*. This result suggests that a drastic change has taken place in the lineage leading to *D. suzukii* in how females respond to chemical cues produced by microbes. We also found that hardness of the substrate affects the response to microbial growth, indicating that mechanosensory stimuli interact with chemosensory invoked decisions to select or avoid oviposition sites.

1. Introduction

Oviposition site selection is a critical factor in determining the survival rate of offspring in insect species. A nutritionally suitable resource may be heavily utilized by other insects and the offspring may suffer from intense competition. The females of *Drosophila suzukii* Matsumura (Diptera: Drosophilidae) have the ability to pierce the skin of ripening fruits and lay eggs into the flesh by using serrated ovipositors [1–3]. Because many other closely related *Drosophila* species lay eggs onto fermented fruits, this behavior allows *D. suzukii* to utilize a carbohydrate-rich resource before competition becomes intense [4,5].

The behavioral shift to deposit eggs into ripening fruits must have been accompanied by changes not only in the ovipositor morphology but also in the sensory systems used to evaluate the oviposition substrate. Karageorgi *et al.* [6] showed that when given the choice between ripe and rotten strawberry fruits, *D. suzukii* strongly preferred ripe over rotten fruit, whereas *D. melanogaster* showed an opposite tendency and preferred rotten fruit, consistent with other studies [7,8]. In the same experiment, *D. biarmipes*, a closely related species of *D. suzukii*, showed no preference between ripe and rotten fruit, indicating that they are at an intermediate evolutionary stage between *D. suzukii* and *D. melanogaster*. It has also been shown in the same study that while *D. biarmipes* and *D. melanogaster* show similarly strong preferences for soft substrates, *D. suzukii* lay eggs onto both hard and soft agarose gel substrates, a pattern similar to other studies [4,9]. Therefore, these studies indicate that *D. suzukii* have widened the range of potential substrates to include those with different degrees of hardness and does not necessarily prefer a harder fruit surface [10–12]. Thus, hardness alone does not account for the strong preference for ripe fruits as an oviposition substrate. Other sensory modifications are

*Authors for correspondence:

Aya Takahashi (ayat@tmu.ac.jp)

Joanne Y. Yew (jyew@hawaii.edu)

13 also likely to underlie the radical shift to an unexploited resource in *D. suzukii* after divergence from the *D.*
14 *biarmipes* lineage.

15 The evolutionary changes in the *D. suzukii* chemosensory system and response to attractants from
16 ripening fruits have been documented [6,13,14], but the chemical properties of possible repellent substances of
17 fermenting fruits have not been investigated in detail. As shown in a previous study, inoculation of the
18 substrate from *D. melanogaster* adults significantly reduced the number of eggs laid by *D. suzukii* [15]. The
19 identity of the aversive substances left by *D. melanogaster* is not known. The deposition of aggregation
20 pheromones is one likely factor [16–18]. Additionally, microbial populations on fermenting fruits originating
21 from the surrounding environment as well as individuals that have visited the fruit, represent another source
22 of aggregation signals. The presence of non-pathogenic microbes guides a wide array of behavioral decisions
23 in insects, including adult aggregation, feeding decisions, and oviposition choice [19–23]. Partnering with
24 commensal microbes provides several benefits for insect hosts including protection from pathogenic microbes,
25 increased access to nutritional resources, and improved offspring survival [24]. The response of *D. suzukii*
26 oviposition to the microbial environment has been largely unstudied and represents an aspect of its social and
27 ecological interactions that may have influenced the new host exploitation in this species.

28 Assessing the fruit condition and making the decision to select the oviposition site involve an
29 integration of multiple sensory cues. It has been shown that *D. suzukii* has the ability to make complex
30 decisions between healthy and fermenting fruits depending on the availability of the resource [8]. In *D.*
31 *melanogaster*, mechanosensory (texture) and chemosensory (taste) information are integrated to direct feeding
32 and oviposition decisions [25–27]. It is an intriguing question as to how different sensory information is
33 processed and integrated in *D. suzukii* in comparison to *D. biarmipes* and *D. melanogaster*, both of which have
34 different decision making criteria for choosing oviposition sites.

35 In this study, we investigate the effects of commensal microbes on oviposition site preferences with
36 independent of and in combination with the effect of the substrate hardness, in *D. suzukii*, *D. biarmipes* and *D.*
37 *melanogaster*. In our assay, *D. suzukii* exhibited a strong avoidance of microbes transferred from other flies.
38 This response was distinct from the other two species suggesting that the behavior has evolved in the lineage
39 leading to *D. suzukii* after the split from *D. biarmipes*. Furthermore, we tested the combinatorial effect of the
40 hardness and the presence or absence of microbes on the oviposition site selection. The mechanical stimuli
41 provided by substrate hardness superseded the influence of microbial chemical signals. We show that this
42 property was conserved among the three species despite differential preference towards hardness and
43 microbial stimuli.

45 2. Materials and Methods

46 2.1. Fly strains

47 The following strains were used to compare the oviposition site preference: *D. suzukii* strain Hilo collected in
48 Hilo, Island of Hawai'i, U.S.A. in 2017, *D. biarmipes* strain MYS118, collected in Mysore, India in 1981, and *D.*
49 *melanogaster* strain Canton S BL#9515. All the strains were maintained at 25 ± 1 °C under the 12 h light: 12 h
50 dark light cycle. All flies were fed with standard corn meal food mixed with yeast, glucose, and agar.

52 2.2. Oviposition assay to test the preference for substrates with microbial growth

53 The procedure is illustrated in figure 1. Inoculation was conducted by using *D. melanogaster* (3 to 7 days after
54 eclosion), *D. biarmipes* (3 to 7 days after eclosion) or *D. suzukii* (7 to 14 days after eclosion). One hundred to 150
55 flies were placed into the inoculation chamber without anesthesia and left for 8 h. An inoculation chamber
56 consists of a plastic cup (100 mL, Tri-Corner Beakers) and a petri dish (57 mm diameter × 16 mm height, IWAKI
57 1010-060) filled with 5 mL 1% agar (*Drosophila* agar type II, Apex) in apple juice (SUNPACK, JAN code:
58 4571247510950) diluted to 50%. No flies were placed into the control inoculation chamber. After inoculation,
59 the surface of the substrate was washed with 1 mL distilled water by pipetting 10 times. Wash solutions (100
60 µL) from inoculated or control plates (figure 1a) were spread onto a new agar plate (40 mm diameter × 13 mm
61 height, Azunol 1-8549-01) and incubated for 24 h at 25 ± 1 °C. Microbial colonies were visible on the media
62 spread with aqueous solution from the inoculated media after 24 h incubation.

63 The oviposition assay was conducted with a petri dish chamber (150 mm diameter × 20 mm height,
64 IWAKI 3030-150) containing four Φ40 mm petri dishes with two types of media placed alternatively (figure 1b).
65 Twenty females and 10 males were placed into the chamber without anesthesia within 3 h before the dark
66 cycle and kept for 16 h in the dark condition. The assay was conducted under the condition of 25 ± 1 °C and 50
67 ± 5% relative humidity. The photo image of each petri dish with substrate was taken by a camera (Olympus
68 DP73) with transmitted light from the bottom. The number of eggs on each substrate was counted.

69 The preference index (PI) for the substrate with microbial growth was calculated by using the
70 following formula:

$$\text{Preference index (PI) for substrate with microbial growth} = \frac{N_{\text{inoculated}} - N_{\text{control}}}{N_{\text{inoculated}} + N_{\text{control}}}$$

where $N_{\text{inoculated}}$ and N_{control} are the total numbers of eggs on the substrates with microbial growth and the control plates, respectively.

To confirm that the PI measurements for substrates inoculated with microbial colonies reflect the activity of microbes, collected solutions from the inoculated media were filter sterilized using a syringe filter (0.22 μm Millex[®]-GV Filter Unit). After washing the surface of the inoculated medium by repeatedly pipetting 1.2 mL distilled water 10 times, the aqueous solution was filtered and used in the oviposition assay as described above.

2.3. Oviposition preference assay for substrate hardness, with and without microbes

Inoculant from *D. melanogaster* was collected from three inoculation chambers, pooled, and divided into 24 (8 \times 3 species) Φ 40 mm petri dishes with medium. Plates without any solution were used for the assays that did not test microbial inoculation. The remaining steps were the same as in 2.2. The PI for the soft substrate was calculated by using the following formula:

$$\text{Preference index (PI) for soft substrate} = \frac{N_{1\% \text{ agar}} - N_{3\% \text{ agar}}}{N_{1\% \text{ agar}} + N_{3\% \text{ agar}}},$$

where $N_{1\% \text{ agar}}$ and $N_{3\% \text{ agar}}$ are the total numbers of eggs on the 1% and 3% agar media, respectively.

2.4. 16S-rRNA gene sequencing of microbial colonies used for the oviposition assays

In order to collect the microbes tested for the oviposition assays, the surface of the inoculated substrate was washed with distilled water as described above. The solution was diluted to 200 μL total volume and spread onto a petri dish (90 mm diameter \times 16 mm height, IWAKI SH90-15) filled with 10 mL apple juice agar as described above. The media were incubated for 24 to 40 hours at 25 ± 1 $^{\circ}\text{C}$ and single colonies were selected randomly for DNA extraction. Each colony was picked with a 10 μL pipette tip, suspended in 20 μL of sterile water, and incubated for 15 min at 95 $^{\circ}\text{C}$ after adding 20 μL 100 mM NaOH. Then, 4.4 μL of 1 M Tris-HCl pH 7.0 was added to each sample and used as template DNA.

Colony PCR was performed with 16S-rRNA universal primers 8F (AGAGTTTGTATCMTGGCTCAG) [28,29] and 1492R (GGYTACCTTGTTACGACTT) [30,31] in a 30 μL reaction using Ex Taq (TaKaRa). Amplification condition for the PCR included an initial denaturation step of 95 $^{\circ}\text{C}$ for 3 min, followed by 35 cycles of 95 $^{\circ}\text{C}$ for 30 s, 53 or 55 $^{\circ}\text{C}$ for 30 s, and 72 $^{\circ}\text{C}$ for 60 s, and a final extension step of 72 $^{\circ}\text{C}$ for 5 min. Reaction products were checked for size and purity on 1% agarose gel and were sequenced after purification by using either BrilliantDye Terminator Cycle Sequencing Kit v2.1 (Nimagen) and a 3130 xl DNA Analyzer (Thermo Fisher Science) or BigDye Terminator v3.1 Cycle Sequencing Kit (Thermo Fisher Science) and a 3170xl DNA Analyzer (Thermo Fisher Science). Sequences were aligned by using MEGA7 [32] and trimmed from the nucleotide positions 61 to 628 of the *Escherichia coli* reference sequence (CP023349.1:226,883-228,438). The genus level identity of each sequence was assigned by the highest score entries in the NCBI database, "16S ribosomal RNA (Bacteria and Archaea type strains)" (as of May 28, 2020) by local BLAST (BLAST+ 2.10.0).

3. Results

The oviposition site preference of *D. suzukii* for ripening fruits relies on shifts in mechanosensation as well as chemosensation [6]. Recent work has shown that consistent with their preference towards ripening fruits over fermenting fruits, *D. suzukii* females tend to lay more eggs on non-inoculated media compared to media inoculated by *D. melanogaster* [15]. Our study focused on determining whether microbial presence and the hardness of the oviposition substrate form the basis of *D. suzukii* oviposition decisions.

3.1. Oviposition site preference against the presence of microbes

Oviposition can be influenced by pheromones or microbial presence. To distinguish between these two possibilities, we first established a method to test only the contribution of microbial growth to oviposition site preference. A water wash was used to collect substances deposited by adult flies and the inoculum was applied to sterile media (figure 1a). Many of the known pheromones used for *Drosophila* chemical communication are hydrophobic hydrocarbons, wax esters and alcohols [33], and are thus, not soluble in water and unlikely to be transferred in the water wash. After incubation, microbial colonies were visible on the inoculated media. Media that had been exposed to water wash from control chambers did not have visible colonies.

The results from the oviposition assay on soft medium (1% agar) indicated that *D. suzukii* avoided oviposition substrates with microbial colonies (figure 2a, Table S1). Given a choice between substrates with

aqueous solutions from inoculated and non-inoculated media, the *D. suzukii* preference index (PI) was significantly less than 0, indicating that the microbial growth discouraged oviposition. By contrast, *D. melanogaster* preferred ovipositing on substrates with microbial growth (figure 2a), indicating that the presence of microbes positively influenced the choice of oviposition site for this species. To trace the evolutionary trajectory of this preference, we also conducted the same experiments using *D. biarmipes*, a closely related species to *D. suzukii*. Remarkably, as with *D. melanogaster*, the microbes positively influenced oviposition site choice of *D. biarmipes* (figure 2a) indicating that the preference for ovipositing at sites with commensal microbes is the ancestral state among these species and that *D. biarmipes* still retain this characteristic. These results were consistent when using microbes from conspecific and heterospecific inoculation (figure 2a). Thus, the drastic change from attraction to avoidance of microbes is predicted to have occurred in the lineage leading to *D. suzukii* after the separation from the *D. biarmipes* lineage.

To confirm that the presence of microbes in the water wash is the primary factor in guiding oviposition, we passed the collected aqueous solution through a 0.22 μm filter to remove microbes and large food particles while keeping nutrients, metabolites, and other small molecules found in feces. In all species, filter-sterilization of the inoculant eliminated both positive and negative oviposition preferences (figure 2b, Table S2). Therefore, microbes that can be removed by a 0.22 μm filter are likely to be the main factor affecting oviposition site preferences.

To identify the main bacterial species that were present in the water washes of inoculated media, we sampled microbial colonies from the medium after 24 h of growth and performed PCR amplification of the 16S-rRNA gene sequence. The bacterial species classified at the genus level and the frequencies estimated from the sampled colonies are shown in figure S1 and Table S3–S5. The bacteria used for our oviposition preference assay were mostly from the *Acetobacter* and *Gluconobacter* genera.

3.2. Combinatorial effect of the presence of microbes and the hardness of the oviposition substrate

In addition to chemosensory signals, another factor that is known to affect *Drosophila* oviposition site preference is the hardness of the substrate. Choice assays using agarose media with different degree of hardness have shown that *D. suzukii* females exhibit a much weaker preference towards softer substrates compared to *D. biarmipes* and *D. melanogaster* [6]. In order to investigate the combinatorial effect of hardness and microbial growth, we conducted choice assays using hard oviposition substrate (3% agar medium) with and without the presence of microbes (figure 2c, Table S6).

When substrates were hard, *D. melanogaster* and *D. biarmipes* showed a PI close to 1, which is indicative of even stronger preferences for ovipositing on media with microbial growth than when using 1% agar media (figure 2a). Interestingly, the aversion to substrates with microbial growth exhibited by *D. suzukii* was reduced when the harder 3% media were used. No significant preference or aversion was detected (figure 2c). From the outcome of this combinatorial assay, it was clear that the hardness of the substrate modifies the preferences against microbes.

Next, we investigated whether the choice between soft (1%) and hard (3%) agar media was affected by the presence of microbes (figure 3a). Our experimental results using 1% and 3% agar media without microbes were consistent with a previous study showing that *D. suzukii* has no or only a slight preference for softer media, in contrast to the strong preference exhibited by the other two species (figure 3b, Table S7).

Interestingly, whether the microbes were present or not did not affect the PI between soft and hard substrates in *D. melanogaster* and *D. suzukii*. The preference towards the softer substrate became significantly weaker when microbes were present than when they were absent in *D. biarmipes*, but only slightly. These results indicate that rather than the presence or absence of microbial growth, the hardness of the substrate is the dominant factor in oviposition site selection.

4. Discussion

4.1. Commensal microbes deposited by flies affect oviposition site preferences in *D. suzukii*, *D. biarmipes*, and *D. melanogaster*, and the preference of *D. suzukii* is distinct from that of the other species

Fruit flies like many other insects coexist with a community of gut microbes, the composition of which can vary to a large extent due to various field and laboratory conditions[34–37]. To elucidate whether gut microbes function as intra- or inter-specific behavioral cues, we examined the influence of fly-deposited microbes on oviposition behavior.

Our results show that egg-laying decisions in *Drosophila* are strongly influenced by the presence of microbial growth, suggesting that microbe-derived cues influence egg-laying decisions in species that use fruit as an oviposition substrate. *D. suzukii* avoided media inoculated with commensal microbes, in contrast to *D. melanogaster* and *D. biarmipes*, both of which showed strong preferences toward microbe-rich media (figure 2). The reversal in preference must have occurred in the *D. suzukii* lineage after the split from *D. biarmipes* consistent with the timing of the host shift to ripening fruits. Therefore, the radical change in microbial preference may have been associated with the new niche exploitation in this lineage.

4.2. Acid producing bacteria differentially affect oviposition behavior amongst *Drosophila* species

The bacterial species used for oviposition preference assays consisted mainly of *Acetobacter* and *Gluconobacter*, both members of the acid-producing Acetobacteraceae family commonly found in the guts of lab-raised and wild fruit fly species [35] including *D. suzukii* [38,39]. Acid-producing bacteria provide benefits for host flies by accelerating growth and offering protection from pathogenic bacteria [40,41]. The colonies grown on the media are not likely to represent the actual composition of fly-associated microbiota in the wild since growth is restricted by diet and the type of media used (agar in apple juice). Flies from natural populations exhibit a more diverse microbiome [36,42]. In addition, our characterization of the microbiome focused only on bacterial species. It is likely that yeast, which are a common symbiont for drosophilids [43], are also part of the inoculum and contribute to oviposition preference [44].

D. melanogaster, *biarmipes*, and *suzukii* exhibited different proportions of *Acetobacter* and *Gluconobacter* (figure S1, Table S3–S5). However, there were no differences in the responses of the three *Drosophila* species to conspecific or heterospecific inoculants, indicating that both *Acetobacter* and *Gluconobacter* have similar effects on the oviposition site choice (figure 2). While *D. suzukii* showed a clear aversion for ovipositing on inoculated media, the response of females to *Gluconobacter* volatiles may be context-dependent. A previous study showed that females starved for 24 h exhibit clear attraction to *Gluconobacter* in an olfactometer bioassay [45]. Taken together with our observation that *D. suzukii* avoids egg-laying in the presence of *Gluconobacter* colonies, it is clear that reproductive and feeding site preferences can be clearly decoupled in this species. Microbial cues that are attractive for feeding may be aversive for oviposition.

4.3. Chemical cues mediating the differential preference against microbes await further investigation

In studies searching for oviposition deterrents for the pest management of fruit crops, at least two chemicals, geosmin and octenol (1-octen-3-ol), both of which are components of volatile metabolites from microorganisms present in rotting fruits, induced aversive responses in *D. suzukii* [46]. However, because these chemicals are known repellents in *D. melanogaster* as well [47,48], the aversion to these microbial compounds is not likely to underlie the *D. suzukii* specific shift in oviposition site.

A study using *D. melanogaster* indicated that female oviposition is guided by sucrose, a gustatory cue used to sense fermentation by lactic acid-producing *Enterococci* bacteria [49]. Interestingly, the olfactory system was shown to be dispensable for ovipositional attraction to these microbes. In contrast, the inhibition of synaptic transmission in sweet sensing gustatory neurons, *Gr5a* and *Gr64a* neurons, impaired the oviposition preference toward fermentation sources. Whether sucrose sensing also mediates the avoidance of acetic acid bacteria in *D. suzukii* would be an intriguing question to pursue. Nevertheless, Silva-Soares et al. [4] showed that *D. suzukii* and *D. biarmipes* have similar oviposition preferences toward sites with a low protein (yeast) to carbohydrate (sucrose) ratio, suggesting that a differential response to sucrose is not likely to explain the contrasting response to acetic acid bacteria products. The microbe-derived chemical cues that govern oviposition response await further investigation.

4.4. Oviposition site hardness supersedes the *D. suzukii* aversion to microbial presence

Integration of different types of stimuli is essential for critical decision-making processes such as the selection of egg deposition sites, a choice that has large influences on the early life performances of the offspring. In *D. melanogaster*, neural circuits governing oviposition site combine information from different modalities [50,51]. Recently, several studies [26,27] elucidated an underlying molecular mechanism for integrating mechanosensory and chemosensory information to make egg-laying decisions in *D. melanogaster*. Our results reveal that two different classes of sensory cues, substrate hardness and the presence of microbes, are integrated in *D. suzukii* oviposition decisions in a manner that is distinct from *D. biarmipes* and *D. melanogaster* (figure 2 and 3). The avoidance of microbes displayed by *D. suzukii* was evident only in the context of a soft substrate (figure 2a) but not a hard one (figure 2c). These results suggest that mechanical cues from surface hardness take precedence over decisions guided by microbial cues. By contrast, the preference exhibited by both *D. melanogaster* and *D. biarmipes* towards microbe-inoculated surfaces strengthened when hard substrates were used (figure 2c), indicating a similar integration of mechanical and microbial chemical cues. Conversely, microbial presence did not affect the choice between hard and soft substrates in all the three species (figure 3).

These results indicate that mechanical and chemical stimuli are not processed additively in these species. The surface hardness modifies the response to microbial cues but not vice versa. It remains to be determined whether surface texture is prioritized in the context of pathogenic microbes. Interestingly, previous studies showed that in female *D. melanogaster*, the presence of chemicals, sucrose and/or fruit juice ingredient obviate the preference for ovipositing on softer surfaces [26,27]. The discrepancy between the

direction of interference between mechanical and chemical stimuli suggests that the hierarchy of cues used in oviposition may depend on the nature of the chemical stimulus.

4.5. The integration of mechanical cues and microbial stimuli is conserved in oviposition choice and reflect differences in ecology

Our findings in this study can be interpreted in the context of natural ecology of *D. suzukii*. In early fruiting season when all the fruits are hard or have no microbial cues, *D. suzukii* females may lay eggs onto any available fruits. This scenario is consistent with the results of our assays using only hard substrate (figure 2c) or only non-inoculated substrates (figure 3). During the ripening period when fruits become softer and ripe, the females may choose fruits with weaker fermentation cues in order to avoid competition with other species, which is consistent with our results using only soft substrate (figure 2a). In late fruiting season when the majority of the fruits are on the ground and rotten, the females may readily lay eggs onto suboptimal fermenting fruits, the situation resembling our assays using only inoculated substrates (figure 3). These explanations are consistent with the study by Kienzle et al. [8], which showed that *D. suzukii* exhibit stronger preferences toward ovipositing in healthy fruits when healthy and fermenting fruits are both abundant compared to when the former are less abundant. The context dependent optimization through seasonal change in host fruit condition might explain the evolutionary background of our findings where substrate hardness takes precedence over microbial presence in the decision to oviposit in this species.

Although surface hardness interacts with the response to commensal microbe cues in *D. biarmipes* and *D. melanogaster* as in *D. suzukii*, there may be some qualitative differences in ecological context between these species. *D. biarmipes* and *D. melanogaster* show a strong preference toward soft substrates inoculated with microbes, and their preferences for microbes is enhanced when the substrate is hard (figure 2). In the field, it may be the case that flies are more likely to use hard fruits in the presence of a microbial signature, which may be indicative of ongoing fermentation. In contrast to *D. suzukii*, both *D. biarmipes* and *D. melanogaster* tend to prefer soft substrates even when all the substrates in the vicinity have microbial growth (figure 3), indicating that mechanical cues supersede microbial presence in oviposition site selection. Therefore, *D. suzukii* may have rapidly adjusted the manner in which mechanical and chemical stimuli are integrated to optimize an egg-laying strategy that is different from other closely related species.

Acknowledgments

We are grateful to K. Nakayama and N. Yoneishi for excellent technical assistance, the UHM Microbial Genetics and Analytical Laboratory for use of facilities, Eurofins Genomics K.K. for Sanger sequencing, and members of the Takahashi lab and Yew lab for helpful discussions.

Funding Statement

This work was supported by JSPS KAKENHI (Grant No. JP19H03276) awarded to A.T.; Department of Defense United States Army Research Office (Grant No. W911NF1610216) and the National Institutes of Health (Grant No. 1P20GM125508) awarded to JYY.

Data Accessibility

The datasets supporting this article have been uploaded as part of the Supplementary Material.

Competing Interests

We have no competing interests.

Authors' Contributions

A.S., J.Y.Y., and A.T. conceived the research and designed the experiments. A.S. performed the experiments. A.S. and K.M.T. analysed the data. A.S. and A.T. drafted the manuscript. All authors gave final approval for publication.

References

- Walsh DB, Bolda MP, Goodhue RE, Dreves AJ, Lee J, Bruck DJ, Walton VM, O'Neal SD, Zalom FG. 2011 *Drosophila suzukii* (Diptera: Drosophilidae): Invasive pest of ripening soft fruit expanding its geographic range and damage potential. *J. Integr. Pest Manag.* **2**, G1–G7. (doi:10.1603/IPM10010)
- Atallah J, Teixeira L, Salazar R, Zaragoza G, Kopp A. 2014 The making of a pest: the evolution of a fruit-penetrating ovipositor in *Drosophila suzukii* and related species. *Proc. R. Soc. London B Biol. Sci.* **281**. (doi:10.1098/rspb.2013.2840)
- Muto L, Kamimura Y, Tanaka KM, Takahashi A. 2018 An innovative ovipositor for niche exploitation impacts genital coevolution between sexes in a fruit-damaging *Drosophila*. *Proc. R. Soc. B Biol. Sci.* **285**, 20181635. (doi:10.1098/rspb.2018.1635)
- Silva-Soares NF, Nogueira-Alves A, Beldade P, Mirth CK. 2017 Adaptation to new nutritional environments: larval performance, foraging decisions, and adult oviposition choices in *Drosophila suzukii*. *BMC Ecol.*, 1–13. (doi:10.1186/s12898-017-0131-2)
- Young Y, Buckiewicz N, Long TAF. 2018 Nutritional geometry and fitness consequences in *Drosophila suzukii*, the Spotted-Wing *Drosophila*. *Ecol. Evol.* **8**, 2842–2851. (doi:10.1002/ece3.3849)
- Karageorgi M, Bräcker LB, Lebreton S, Minervino C, Cavey M, Siju KP, Grunwald Kadow IC, Gompel N, Prud'homme B. 2017 Evolution of multiple sensory systems drives novel egg-laying behavior in the fruit pest *Drosophila suzukii*. *Curr. Biol.* **27**, 847–853. (doi:10.1016/j.cub.2017.01.055)
- Lee JC, Bruck DJ, Curry H, Edwards D, Haviland DR, Van Steenwyk RA, Yorgey BM. 2011 The susceptibility of small fruits and cherries to the spotted-wing *Drosophila*, *Drosophila suzukii*. *Pest Manag. Sci.* **67**, 1358–1367. (doi:10.1002/ps.2225)
- Kienzle R, Groß LB, Caughman S, Rohlf M. 2020 Resource use by individual *Drosophila*

- 1
2
3
4
5
6
7
8
9
10
11
12
13
14
15
16
17
18
19
20
21
22
23
24
25
26
27
28
29
30
31
32
33
34
35
36
37
38
39
40
41
42
43
44
45
46
47
48
49
50
51
52
53
54
55
56
57
58
59
60
9. *suzukiireveals a flexible preference for oviposition into healthy fruits. Sci. Rep.* **10**, 3132. (doi:10.1038/s41598-020-59595-y)
10. Guo L, Zhou Z-D, Mao F, Fan X-Y, Liu G-Y, Huang J, Qiao X-M. 2020 Identification of potential mechanosensitive ion channels involved in texture discrimination during *Drosophila suzukii* egg-laying behaviour. *Insect Mol. Biol.* (doi:10.1111/imb.12654)
11. Burrack HJ, Fernandez GE, Spivey T, Kraus DA. 2013 Variation in selection and utilization of host crops in the field and laboratory by *Drosophila suzukii* Matsumura (Diptera: Drosophilidae), an invasive frugivore. *Pest Manag. Sci.* **69**, 1173–1180. (doi:10.1002/ps.3489)
12. Kinjo H, Kunimi Y, Ban T, Nakai M. 2013 Oviposition Efficacy of *Drosophila suzukii* (Diptera: Drosophilidae) on Different cultivars of blueberry. *J. Econ. Entomol.* **106**, 1767–1771. (doi:10.1603/ec12505)
13. Lee JC, Dalton DT, Swoboda-Bhattarai KA, Bruck DJ, Burrack HJ, Strik BC, Woltz JM, Walton VM. 2016 Characterization and manipulation of fruit susceptibility to *Drosophila suzukii*. *J. Pest Sci.* (2004). **89**, 771–780. (doi:10.1007/s10340-015-0692-9)
14. Keesey IW, Knaden M, Hansson BS. 2015 Olfactory specialization in *Drosophila suzukii* supports an ecological shift in host preference from rotten to fresh fruit. *J. Chem. Ecol.* **41**, 121–128. (doi:10.1007/s10886-015-0544-3)
15. Revadi S *et al.* 2015 Olfactory responses of *Drosophila suzukii* females to host plant volatiles. *Physiol. Entomol.* **40**, 54–64. (doi:10.1111/phen.12088)
16. Shaw B, Brain P, Wijnen H, Fountain MT. 2018 Reducing *Drosophila suzukii* emergence through inter-species competition. *Pest Manag. Sci.* **74**, 1466–1471. (doi:10.1002/ps.4836)
17. Lin C-C, Prokop-Prigge KA, Preti G, Potter CJ. 2015 Food odors trigger *Drosophila* males to deposit a pheromone that guides aggregation and female oviposition decisions. *Elife* **4**, 1–26. (doi:10.7554/elife.08688)
18. Xu P, Atkinson R, Jones DNM, Smith DP. 2005 *Drosophila* OBP LUSH is required for activity of pheromone-sensitive neurons. *Neuron* **45**, 193–200. (doi:10.1016/j.neuron.2004.12.031)
19. Tait G *et al.* 2020 Reproductive site selection: evidence of an oviposition cue in a highly adaptive dipteran, *Drosophila suzukii* (Diptera: Drosophilidae). *Environ. Entomol.* **49**, 355–363. (doi:10.1093/ee/ivaa005)
20. Lewis Z, Lizé A. 2015 Insect behaviour and the microbiome. *Curr. Opin. insect Sci.* **9**, 86–90. (doi:10.1016/j.cois.2015.03.003)
21. Fischer CN, Trautman EP, Crawford JM, Stabb E V, Handelsman J, Broderick NA. 2017 Metabolite exchange between microbiome members produces compounds that influence *Drosophila* behavior. *Elife* **6**. (doi:10.7554/elife.18855)
22. Wong AC-N, Wang Q-P, Morimoto J, Senior AM, Lihoreau M, Neely GG, Simpson SJ, Ponton F. 2017 Gut microbiota modifies olfactory-guided microbial preferences and foraging decisions in *Drosophila*. *Curr. Biol.* **27**, 2397–2404.e4. (doi:10.1016/j.cub.2017.07.022)
23. Jose PA, Ben-Yosef M, Jurkevitch E, Yuval B. 2019 Symbiotic bacteria affect oviposition behavior in the olive fruit fly *Bactrocera oleae*. *J. Insect Physiol.* **117**, 103917. (doi:10.1016/j.jinsphys.2019.103917)
24. Douglas AE. 2015 Multiorganismal insects: diversity and function of resident microorganisms. *Annu. Rev. Entomol.* **60**, 17–34. (doi:10.1146/annurev-ento-010814-020822)
25. Jeong YT, Oh SM, Shim J, Seo JT, Kwon JY, Moon SJ. 2016 Mechanosensory neurons control sweet sensing in *Drosophila*. *Nat. Commun.* **7**, 1–9. (doi:10.1038/ncomms12872)
26. Wu SF, Ja YL, Zhang YJ, Yang CH. 2019 Sweet neurons inhibit texture discrimination by signaling TMC-expressing mechanosensitive neurons in *Drosophila*. *Elife* **8**, 1–24. (doi:10.7554/elife.46165)
27. Zhang L, Yu J, Guo X, Wei J, Liu T, Zhang W. 2020 Parallel mechanosensory pathways direct oviposition decision-making in *Drosophila*. *Curr. Biol.*, 1–14. (doi:10.1016/j.cub.2020.05.076)
28. Weisburg WG, Barns SM, Pelletier DA, Lane DJ. 1991 16S ribosomal DNA amplification for phylogenetic study. *J. Bacteriol.* **173**, 697–703. (doi:10.1128/jb.173.2.697-703.1991)
29. Turner S, Pryer KM, Miao VP, Palmer JD. 1999 Investigating deep phylogenetic relationships among cyanobacteria and plastids by small subunit rRNA sequence analysis. *J. Eukaryot. Microbiol.* **46**, 327–338. (doi:10.1111/j.1550-7408.1999.tb04612.x)
30. Edwards U, Rogall T, Blöcker H, Emde M, Böttger EC. 1989 Isolation and direct complete nucleotide determination of entire genes. Characterization of a gene coding for 16S ribosomal RNA. *Nucleic Acids Res.* **17**, 7843–7853. (doi:10.1093/nar/17.19.7843)
31. Loy A, Lehner A, Lee N, Adamczyk J, Meier H, Ernst J, Schleifer K-H, Wagner M. 2002 Oligonucleotide microarray for 16S rRNA gene-based detection of all recognized lineages of sulfate-reducing prokaryotes in the environment. *Appl. Environ. Microbiol.* **68**, 5064–5081. (doi:10.1128/aem.68.10.5064-5081.2002)
32. Kumar S, Stecher G, Tamura K. 2016 MEGA7: Molecular Evolutionary Genetics Analysis version 7.0 for bigger datasets. *Mol. Biol. Evol.* **33**, 1870–1874. (doi:10.1093/molbev/msw054)
33. Yew JY, Chung H. 2017 *Drosophila* as a holistic model for insect pheromone signaling and processing. *Curr. Opin. insect Sci.* **24**, 15–20. (doi:10.1016/j.cois.2017.09.003)
34. Bing X, Gerlach J, Loeb G, Buchon N. 2018 Nutrient-Dependent Impact of Microbes on *Drosophila suzukii* development. *MBio* **9**, e02199-17. (doi:10.1128/mBio.02199-17)
35. Broderick NA, Lemaitre B. 2012 Gut-associated microbes of *Drosophila melanogaster*. *Gut Microbes* **3**. (doi:10.4161/gmic.19896)
36. Chandler JA, Lang J, Bhatnagar S, Eisen JA, Kopp A. 2011 Bacterial communities of diverse *Drosophila* species: Ecological context of a host-microbe model system. *PLoS Genet.* **7**. (doi:10.1371/journal.pgen.1002272)
37. Wong AC-N, Chaston JM, Douglas AE. 2013 The inconstant gut microbiota of *Drosophila* species revealed by 16S rRNA gene analysis. *ISME J.* **7**, 1922–1932. (doi:10.1038/ismej.2013.86)
38. Chandler JA, James PM, Jospin G, Lang JM. 2014 The bacterial communities of *Drosophila suzukii* collected from undamaged cherries. *PeerJ* **2014**, 1–10. (doi:10.7717/peerj.474)
39. Martínez-Sañudo I, Simonato M, Squartini A, Mori N, Marri L, Mazzon L. 2018 Metagenomic analysis reveals changes of the *Drosophila suzukii* microbiota in the newly colonized regions. *Insect Sci.* **25**, 833–846. (doi:10.1111/1744-7917.12458)
40. Crotti E *et al.* 2010 Acetic acid bacteria, newly emerging symbionts of insects. *Appl. Environ. Microbiol.* **76**, 6963–6970. (doi:10.1128/AEM.01336-10)
41. Shin SC, Kim S-H, You H, Kim B, Kim AC, Lee K-A, Yoon J-H, Ryu J-H, Lee W-J. 2011 *Drosophila* microbiome modulates host developmental and metabolic homeostasis via insulin signaling. *Science* **334**, 670–674. (doi:10.1126/science.1212782)
42. Staubach F, Baines JF, Künzel S, Bik EM, Petrov DA. 2013 Host species and environmental effects on bacterial communities associated with *Drosophila* in the laboratory and in the natural environment. *PLoS One* **8**, e70749. (doi:10.1371/journal.pone.0070749)
43. Stefanini I. 2018 Yeast-insect associations: It takes guts. *Yeast* **35**, 315–330. (doi:10.1002/yea.3309)
44. Bellutti N, Gallmetzer A, Innerebner G, Schmidt S, Zelger R, Koschier EH. 2018 Dietary yeast affects preference and performance in *Drosophila suzukii*. *J. Pest Sci.* (2004). **91**, 651–660. (doi:10.1007/s10340-017-0932-2)
45. Mazzetto F, Gonella E, Crotti E, Vacchini V, Syropas M, Pontini M, Manginckx S, Daffonchio D, Alma A. 2016 Olfactory attraction of *Drosophila suzukii* by symbiotic acetic acid bacteria. *J. Pest Sci.* (2004). **89**, 783–792. (doi:10.1007/s10340-016-0754-7)
46. Wallingford AK, Hesler SP, Cha DH, Loeb GM. 2016 Behavioral response of spotted-wing *Drosophila suzukii* Matsumura, to aversive odors and a potential oviposition deterrent in the field. *Pest Manag. Sci.* **72**, 701–706. (doi:10.1002/ps.4040)
47. Knaden M, Strutz A, Ahsan J, Sachse S, Hansson BS. 2012 Spatial representation of odorant valence in an insect brain. *Cell Rep.* **1**, 392–399. (doi:10.1016/j.celrep.2012.03.002)
48. Stensmyr MC *et al.* 2012 A conserved dedicated olfactory circuit for detecting harmful microbes in *Drosophila*. *Cell* **151**, 1345–1357. (doi:10.1016/j.cell.2012.09.046)
49. Liu W, Zhang K, Li Y, Su W, Hu K, Jin S. 2017 Enterococci mediate the oviposition preference of *Drosophila melanogaster* through sucrose catabolism. *Sci. Rep.* **7**, 1–14. (doi:10.1038/s41598-017-13705-5)
50. Yang CH, Belawat P, Hafen E, Jan LY, Jan YN. 2008 *Drosophila* egg-laying site selection as a system to study simple decision-making processes. *Science* **319**, 1679–1683. (doi:10.1126/science.1151842)
51. Joseph RM, Devineni A V., King IFG, Heberlein U. 2009 Oviposition preference for and positional avoidance of acetic acid provide a model for competing behavioral drives in *Drosophila*. *Proc. Natl. Acad. Sci. U. S. A.* **106**, 11352–11357. (doi:10.1073/pnas.0901419106)

Figure captions

Figure 1. Experimental scheme of the oviposition assay to quantify response to water-soluble substances deposited by flies on the surface of media. (a) Water-soluble substances are collected from inoculated and control plates. (b) Oviposition assay using media inoculated with solutions from (a) for 24 h.

Figure 2. Comparisons of the preference indices (PIs) of *D. melanogaster*, *D. biarmipes*, and *D. suzukii* for oviposition substrates treated with inoculant from conspecific (open boxplots) or heterospecific (filled boxplots in gray) flies. (a) The PIs assayed on soft substrate (1% agar medium) with and without inoculant treatment (microbial growth). (b) The PIs assayed on 1% agar medium for substrates treated with sterile filtered solutions of inoculant. (c) The PIs assayed on hard oviposition substrate (3% agar medium) with and without inoculant treatment (microbial growth). Control substrates were treated with solutions from non-exposed (non-inoculated) substrate in all assays. Species used for heterospecific inoculations were conducted using *D. suzukii* for *D. melanogaster* assay, and *D. melanogaster* for *D. biarmipes* and *D. suzukii* assays. Results from assays with fewer than 10 eggs on either substrate were excluded from the analysis. Box signifies the upper and lower quartiles and horizontal bar indicates median. Upper and lower whiskers represent maximum and minimum $1.5 \times$ interquartile range, respectively. The difference from PI = 0 (no preferences) was tested by Wilcoxon signed rank test with Bonferroni correction for multiple comparisons (6 tests). *: $p < 0.05$, ns: $p \geq 0.05$.

Figure 3. Preference indices (PIs) for the soft substrate with and without microbes. (a) The substrate placement in the chambers for the oviposition assay. "1%" and "3%" indicate soft (1% agar medium) and hard (3% agar medium) oviposition substrates, respectively. The microbe (+) chambers have been treated with inoculant collected from substrate surface exposed to *D. melanogaster*; microbial (-) chambers were treated with inoculant from non-exposed surfaces. (b) The preference indices (PI) for soft oviposition substrate in the absence (open boxplots) and presence (filled boxplots in gray) of microbes. Results from assays with fewer than 10 eggs on either substrate were excluded from the analysis. Box signifies the upper and lower quartiles and horizontal bar indicates median. Upper and lower whiskers represent maximum and minimum $1.5 \times$ interquartile range, respectively. Statistical significance was tested by permutation test with Bonferroni correction for multiple comparisons (6 tests). *: $p < 0.05$, ns: $p \geq 0.05$.

(a)

(b)

Appendix B

Response to reviewers' comments

We thank Prof. Sprecher for handling our paper and the reviewers for their constructive comments, which have helped improve our manuscript substantially.

Associate Editor Comments to Author (Professor Simon Sprecher):

Associate Editor: 1

Comments to the Author:

The reviewers are overall positive about the manuscript, but raise a few points that have to be addressed.

We have responded to each point addressed by the reviewers below. For the comments from Reviewer 1 marked on the pdf, we have replied directly to them on the pdf manuscript. Page and line numbers below are of the revision-tracked version of our manuscript.

Reviewer comments to Author:

Reviewer: 1

Comments to the Author(s)

I think this article makes an interesting contribution to the ecology of *Drosophila suzukii*. I have the following points of criticism which should be addressed to give the article the necessary attention by readers. I have made specific remarks as comments directly in the pdf document, so my comments here are only general:

We thank the reviewer for the criticisms and have replied to each comment marked on the pdf document by writing a reply under the comment in the same markup balloon.

1. it would help for the overall understanding if the use of terms and procedures were better justified. In particular, one can only guess why 'substrate hardness' is so important in this article and what the methodological approach actually investigates. In my view, a general research question is also missing.

We agree that the reason we tested the effects of substrate hardness was not clearly stated. We have added a phrase in the abstract (p.1, line 8) and in section 3.2 (p.4, line 160) to clarify this point. Regarding the research question, we have extensively revised the section 1 (p.2, line 31-36) to put forward a hypothesis and guide the readers to the main question.

2. several conclusions are in my opinion not allowed or too strong (see my comments in the pdf document). And here also the problem arises that it is not clear why substrate hardness should be so central, this will only be clarified in the discussion. Personally I also think that this discussion of substrate hardness is only incomplete. I miss the point in the discussion that *D. melanogaster* and others need injured fruits, they can also be unripe and of high substrate hardness, ok, then they might not be so super attractive but accessible. But ripe and internally soft fruits are irrelevant for *D. melanogaster*, because there is no wounded site; and *D. suzukii* cannot perceive the soft interior of a ripe fruit by only having access to the fruit skin.

Indeed, egg laying decisions depend on whether the fruit is injured or not, as we have shown in our paper, Kienzle et al. 2020. The discussion would benefit if this aspect of the fruit injury, which results in a 'soft' substrate, were to be compared to the manipulated substrate hardness with agar.

The conclusions that were marked up to be too strong in the pdf document have been weakened (see our replies in the markup balloons). Regarding the hardness issue, we have added a new paragraph in section 4.5 (p.6, lines 271-275) to raise the point that the agar media used in our assay have uniform texture and may not completely reflect the condition of the real ripening fruits with partially damaged skin in the field. We appreciate the reviewer for raising the issue since we think that the paper benefits substantially from the discussion. We have also modified the phrases in the following paragraphs (p.6, lines 281, 283, 296-297) to incorporate the point about the fruit skin damage.

Reviewer: 2

Comments to the Author(s)

This is a study that contributes to a better understanding of the biology of several *Drosophilid* fly species, together with the role of fruit firmness and presence of micro-organisms.

The authors did a good job to conduct rigorous science and clearly described their results. The publication should be accepted with minor revisions.

We thank the reviewer for the encouraging comment.

The authors are encouraged to describe why the oviposition arenas were not ventilated, and what the shortcomings of this experiment were. With ventilation, a volatile cloud is removed and will likely contribute to more focused selection of certain sites that either contain, or do not contain the studied microorganisms.

We thank the reviewer for raising this issue. The oviposition arena design was based on a previous article, which also used a system without ventilation (Oviposition preference for and positional avoidance of acetic acid provide a model for competing behavioral drives in *Drosophila*. (Ryan M. Joseph, Anita V. Devineni, Ian F. G. King, Ulrike Heberlein. Proceedings of the National Academy of Sciences Jul 2009, 106 (27) 11352-11357; DOI: 10.1073/pnas.0901419106).

We realize now that lack of ventilation may cause a bias due to the buildup of a volatile cloud, which may 1) unfairly bias the experiment because it creates an unnaturally concentrated cue and biases fly preferences in a way that is different from natural conditions, or 2) obscure site preference, because it creates a homogenous cloud throughout the whole arena. However, such possible bias did not appear to be a substantial factor in our study. Regarding the first possibility, there were instances where the flies showed no preference for microbes, indicating that even if there is a concentrated cloud, there are scenarios where flies ignore the volatiles and make decisions based on other cues like substrate hardness (see *D. suzukii* in Fig 2c). For the second possibility, there are instances where a clear

choice for inoculated substrates was made between the media placed within a chamber (see Fig 2a, *D. melanogaster* and *D. biaramipes* in Fig 2c, and *D. melanogaster* and *D. biaramipes* in Fig 3b), indicating that even if there is a cloud of volatiles throughout the arena, there is still enough difference in the signals coming from inoculated substrates to influence choice. Nevertheless, it may be the case that the lack of ventilation will not reflect behavior in the wild. We have added a paragraph in section 4.3 (p.5, line 237 – p.6, line 245) to address this potential shortcoming.

The authors should cite work (2 publications of Ioriatti et al., there are several others that are also somewhat related on winegrape, which describe the interactions of Drosophilids as vectors of several of the microorganisms mentioned in this paper, it will strengthen their arguments and evidence of scholarship.

We appreciate the suggestion and cited Ioriatti et al. 2015 and 2018 in section 4.2 (p.5, lines 202-205). These papers mention about *Gluconobacter* and *Acetobacter* found on our inoculated media, therefore, we agree that the literatures are highly relevant to our study.

Although this is not the focus of the current paper, additional discussion of volatiles emanating from the microorganisms may be important as well.

We have added this statement in section 4.3 (p.5, line 237 – p.6, line 245) to address this point.